# Contrast-Enhanced Computed Tomography and Laboratory Parameters as Non-Invasive Diagnostic Markers of Pancreatic Fibrosis

**DOI:** 10.3390/diagnostics13142435

**Published:** 2023-07-21

**Authors:** Igor E. Khatkov, Dmitry S. Bordin, Konstantin A. Lesko, Elena A. Dubtsova, Nikolay S. Karnaukhov, Maria A. Kiriukova, Nadezhda V. Makarenko, Alexey S. Dorofeev, Irina V. Savina, Diana A. Salimgereeva, Elena I. Shurygina, Ludmila V. Vinokurova

**Affiliations:** 1A.S. Loginov Moscow Clinical Scientific Center, 111123 Moscow, Russiad.bordin@mknc.ru (D.S.B.); e.dubtsova@mknc.ru (E.A.D.); n.karnaukhov@mknc.ru (N.S.K.); kiryukovam@yandex.ru (M.A.K.); n.makarenko@mknc.ru (N.V.M.); a.dorofeev@mknc.ru (A.S.D.); i.savina@mknc.ru (I.V.S.); d.salimgereeva@mknc.ru (D.A.S.); e.shurygina@mknc.ru (E.I.S.); l.vinokurova@mknc.ru (L.V.V.); 2Chair of Faculty Surgery No. 2, A.I. Yevdokimov Moscow State University of Medicine and Dentistry, 127473 Moscow, Russia; 3Chair of General Medical Practice and Family Medicine, Tver State Medical University, 170100 Tver, Russia

**Keywords:** chronic pancreatitis, pancreatic fibrosis, computed tomography, attenuation, fibronectin, hyaluronic acid

## Abstract

Pancreatic fibrosis (PF) is a part of the pathogenesis in most pancreatic disorders and plays a crucial role in chronic pancreatitis development. The aim of our study was to investigate a relationship between PF grade and signs in resected pancreatic specimens, and the results of both multidetector computed tomography (MDCT) post-processing parameters and fibronectin (FN), hyaluronic acid (HA), matrix metalloproteinase (MMP)-1, and MMP-9 serum levels. The examination results of 74 patients were analyzed. The unenhanced pancreas density (UPD) value and contrast enhancement ratio (CER) showed statistically significant differences in groups with peri- and intralobular fibrosis grades, an integrative index of fibrosis, inflammation in pancreatic tissue, and pancreatic duct epithelium metaplasia, while the normalized contrast enhancement ratio in the venous phase (NCER VP) significantly differed with the perilobular fibrosis grade, integrative fibrosis index, and inflammation (*p* < 0.05). The blood FN level showed a weak positive correlation with the intralobular fibrosis grade (rho = 0.32, *p* = 0.008). The blood level of HA positively correlated with the presence of prominent and enlarged peripheral nerves (rho = 0.28, *p* = 0.02) and negatively correlated with the unenhanced pancreas density value (rho = −0.42, *p* = 0.0001). MMP-1 and MMP-9 values’ intergroup analysis and correlation did not show any statistical significance. The UPD value, NCER VP, and CER, as well as blood levels of FN and HA, could be used in non-invasive PF diagnosis.

## 1. Introduction

Chronic pancreatitis (CP) and pancreatic cancer (PC) are major gastrointestinal pancreatic disorders encountered worldwide. CP is a digestive disorder with severe complications in the end stage, such as exocrine and endocrine insufficiency, and it is a risk factor for pancreatic ductal adenocarcinoma. In Europe, the prevalence of CP is 120 per 100,000 persons, which places a significant strain on healthcare systems [1]. Recent guidelines focus on early diagnosis of CP to avoid late-stage complications and improve clinical outcomes, diagnosis, and treatment before CP becomes established and irreversible [2,3]. The importance of early CP diagnosis has been growing tremendously, especially considering the fact that CP is a risk factor of pancreatic ductal adenocarcinoma [4]. PC ranks fourteenth in cancer incidence and is the seventh highest cause of cancer mortality in the world. Its global burden has more than doubled over the past 25 years [5,6]. PC prognosis is among the worst of all common human tumors and remains a challenge for both clinical medicine and research. The five-year survival rate is 2–9%. [7,8]. Therefore, better and earlier CP diagnosis might improve prognosis and outcomes in PC patients.

According to the mechanistic definition of CP, the disease is a pathologic fibro-inflammatory syndrome of the pancreas in individuals with genetic, environmental, and/or other risk factors who develop persistent pathologic responses to parenchymal injury or stress. This definition excludes autoimmune inflammation, inflammation, and fibrosis arising from the islets related to long-standing diabetes mellitus, age-related atrophy or fibrosis, inflammation upstream of a duct-obstructing mass, etc. [3,9]. The American College of Gastroenterology recommends multidetector computed tomography (MDCT) or magnetic resonance imaging (MRI) as the first-line diagnostic modality in CP. According to these guidelines, endoscopic ultrasonography (EUS), due to its invasiveness and lack of specificity, should be performed only if the diagnosis is still in question after cross-sectional imaging. Additionally, CP via routine MDCT and MRI is diagnosed at a late stage with irreversible morphological damage and variable clinical manifestations [9]. Nowadays, MDCT has become the most widely used diagnostic technique not only for CP but also for CP and PC differential diagnosis [10].

MDCT is still considered a non-invasive diagnostic tool for detecting pancreatic fibrosis [11]. There are data on predicting postoperative pancreatic fistulas by number or micro-calcifications depicted by CT scan [12,13]. Obviously, calcification count is, though prominent, a belated parameter for an early pancreatic fibrosis assessment. However, with the main feature as a calcification count higher than 10, MDCT was suggested as an image prediction tool for severe pancreatic fibrosis [14].

MDCT still has its limitations. Among them are radiation exposure and failure to detect early pancreatic fibrosis. MRI, despite providing a high-quality visualization of the pancreas, especially the pancreatic ductal system, cannot distinguish the subtle morphological changes of parenchyma. Endoscopic strain and shear wave elastography are also available as alternative techniques for fibrosis assessment. However, both of them are operator-dependent invasive procedures and offer a point-targeted evaluation [11]. Therefore, no current radiology technique is precise enough to be used alone as a non-invasive modality for early CP and fibrotic change diagnosis.

As permanent and disorganized pancreatic fibrosis is thought to result from an imbalance between the synthesis and degradation of extracellular matrix (ECM) proteins, the products of the latter could serve as early biomarkers of fibrotic changes [9]. There are biomarkers that could be of use in pancreatic fibrosis detection, although the vast majority of them focus on PC diagnosis [15]. The choice of the four most prominent in the first stage of our study was conditioned by the activity of these biomarkers in pancreatic diseases’ development, including their potential role in CP progress [16,17,18].

Matrix metalloproteinases (MMPs) are a family of zinc-containing zymogen endopeptidases that are structurally related and degrade a wide range of ECM components: MMP-1 (interstitial collagenase) has the most prominent activity against collagen types I, II, and III, while MMP-9 (gelatinase B) is most active against collagen types IV and V, as well as gelatin. MMPs were demonstrated to be involved in pancreatic fibrosis pathogenesis [19,20,21]. Previously, CP patients were shown to have higher plasma concentrations of MMP-9 than healthy individuals [18]. In contrast, MMP-1 levels were shown to be lower in a rodent model with induced CP [22].

Another biomarker is hyaluronic acid (HA). It is present in almost every tissue in the human body. HA is a known biomarker of liver fibrosis and successfully used in the diagnosis of different liver diseases associated with fibrosis. HA is a chief component of ECM in connective tissues, but it is also found in the pericellular and intracellular matrix [23,24]. Moreover, it is a major component of the pancreatic cancer microenvironment and richly abundant in pancreatic tumors [25].

Fibronectin (FN), as a provisional part of ECM, is heavily present in PC stroma but not in normal tissues, supporting its metastatic spread and chemo resistance, as well as neoangiogenesis [16,26]. Recently, a growth in the FN level was shown in experimental pancreatic fibrosis [27].

The CP pathophysiology is complex, and all existing theories converge on the same outcome characterized by a progressive irreversible loss of functional pancreas parenchyma and its replacement with fibrotic tissue that eventually results in exocrine and endocrine insufficiency of the organ. Given the high variability of CP clinical representation, histopathological evidence of CP as well as pancreatic fibrosis in its stem is often needed [28]. Early diagnosis of pancreatic fibrosis gives an invaluable opportunity to be a part of CP staging as well as preventing disease progression. Current cross-sectional techniques and biomarkers are not elaborate enough for early and precise CP diagnosis, which essentially limits its non-invasive diagnostics [3].

Nevertheless, accurate diagnosis of pancreatic fibrosis is possible via histopathology examination only. One way of collecting the specimen is via pancreatic biopsy, which is a difficult procedure with a relatively high complication rate [29]. The majority of patients with CP and, presumably, pancreatic fibrosis, however, do not show indications for a biopsy. Another option might be studying tissue obtained during draining surgeries such as lateral pancreaticojejunostomy. These, however, are not of such common use due to the satisfactory results of CP conservative treatment and the rarity and complexity of the procedure [30].

Obtaining specimens for fibrosis analysis represents an issue, as deliberate tissue collection must be conducted only when medically indicated. Given that non-invasive approaches are preferable and available data on pancreatic fibrosis assessment in the published literature are limited, we planned to assess pancreatic fibrosis in patients that underwent preplanned pancreatic surgery for benign and malignant pancreatic lesions and clarify the interconnection between pancreatic fibrosis grade, MDCT parameters, and potential pancreatic fibrosis serum biomarkers.

## 2. Materials and Methods

### 2.1. Patient Data

We conducted an observational single-center prospective study in a high-volume tertiary care center. The protocol of the study was registered at ClinicalTrials.gov, accessed on 25 June 2023 (NCT05775107). From April 2022 to March 2023, we enrolled 74 adult patients who were preplanned to undergo a pancreatic resection for either CP or PC. Exclusion criteria were an unresectable pancreatic tumor; not signed voluntary informed consent due to mental disorder or severe clinical conditions; and inaccessibility of clinical, radiological, biochemical, and morphological data. All patients underwent contrast-enhanced 128-row abdominal MDCT examinations and serum biomarkers assessment at least 3 days before surgery. There were no additional diagnostic procedures exceeding routine diagnostic examination. The final diagnoses were confirmed postoperatively by histopathological examination of the surgical specimens. We prospectively collected CT scan parameters, serum biomarker levels, and histopathology reports of the resected pancreatic parenchyma specimens and stored them in an electronic database that was retrospectively analyzed.

### 2.2. MDCT Examination

We used a 128-row multidetector helical CT scanner (Aquilion CXL 128, Toshiba, Tokyo, Japan) and a non-ionic iodine contrast agent with an iodine concentration of 350 mg/mL for contrast enhancement (Omnipaque, GE healthcare, Fairfield, CT, USA; total of 100 mL). The CT scanner parameters used were 120 kV and 240 mA, allowing for variation according to body habitus. The contrast injection rate was 3.5–5 mL/s via the cubital vein with a mechanical power injector. All the CT scans included precontrast, arterial (30 s), pancreatic (45–50 s), and portal venous (60–80 s), as well as delay (8 min) enhancement phases. Axial images with a slice thickness of 1 mm were obtained.

### 2.3. MDCT Result Post-Processing

All MDCT scans in DICOM files were analyzed by an experienced radiologist specialized in abdominal radiology. We calculated the normalized contrast enhancement ratio (NCER) during the pancreatic (PP) and venous phases (VP) as described by Torphy et al. [31], as well as the contrast enhancement ratio (CER) between the VP and non-contrast MDCT according to the technique described by Hashimoto et al. [32]. The formulae are presented in Table 1.

We measured the attenuation value and tissue density in the pancreatic tissue area, sized 0.2–0.8 cm^2^, close to the pancreatic lesion that was preplanned to be resected along with the lesion during surgery (Figure 1). Only these tumor-free tissue samples underwent subsequent histopathology analysis. In patients with CP, we measured the attenuation value and tissue density in the pancreatic tissue area localized in the part that would be resected during drainage surgery (Figure 2).

### 2.4. Histopathology Analysis

The specimens of pancreatic tissue with a tumor surrounded by intact pancreatic tissue resected during pancreatic surgery were studied by two experienced pathologists using a histological method and light microscopy. Only samples of intact pancreatic tissue beyond the tumor were analyzed (Figure 3).

The material was preprocessed in a Leica ASP6025S automatic histological processor (Leica Biosystems, Wetzlar, Germany). Sections of pancreatic tissue with a thickness of 4.0 μm were prepared for examination on a Leica RM 2125 RTS rotary microtome (Leica Biosystems, Wetzlar, Germany) using histological staining (Mayer’s hematoxylin and eosin staining) via a Leica ST5010 AXL machine (Leica Biosystems, Wetzlar, Germany). Morphometry and microphotography of histological samples were performed on an Olympus light optical microscope BX51 Multihead 10 Headed Teaching System with 2X Objective/Pathology (Olympus, Beijing, China).

The degree of fibrosis of the pancreatic tissue was assessed by a semi-quantitative method in preparations stained with Mayer’s hematoxylin and eosin using the rating scale by Kloppel and Maillet (Table 2) [33]. The scoring system distinguishes focal from diffuse and perilobular from intralobular types of fibrosis, with the final total fibrosis score ranging from 1 to 12, including the degree of intralobular and perilobular fibrosis and their integrative index. It was also recommended by the working group for the international consensus guidelines for chronic pancreatitis in collaboration with the International Association of Pancreatology, American Pancreatic Association, Japan Pancreas Society, and European Pancreatic Club [28].

Pancreatic tissue morphometry included evaluation of pancreatic fibrosis and early CP signs, such as features of inflammation, prominent and enlarged peripheral nerves, pancreatic duct epithelium metaplasia, and presence of protein plugs in pancreatic ducts. The percentage of collagen fibers was assessed using NIS Elements software F4.00.06.Ink (2013). The area of the digital micrograph corresponded to 1 field of view of the microscope at ×300 magnification. Counting was performed in 10 fields of view.

### 2.5. Biomarkers Assessment

Levels of FN, HA, and MMP-1 and MMP-9 in blood serum were determined by enzyme immunoassay using a commercial kit of Technozym Fibronectin reagents (Technoclone, Vienna, Austria), Corgenix hyaluronic acid reagents (Corgenix Headquarters, Broomfield, CO, USA), and RayBio human MMP-1 and MMP-9 reagents (RayBiotech, Peachtree Corners, GA, USA), respectively. The reference intervals, according to the manufacturers, were 70–148 μg/mL for FN, 28.5–75 ng/mL for HA, 88–106 ng/mL for MMP-1, and 84–103 ng/mL for MMP-9. Data registration of all assessed biomarkers was carried out on a Sunrise semi-automatic analyzer (Tecan, Grödig, Austria).

### 2.6. Statistical Analysis

We assessed intergroup differences between all MDCT post-processing values including pancreatic density in non-enhanced images, PP and VP, NCER PP, NCER VP, and CER; and mean values of FN, HA, MMP-1, and MMP-9 in groups divided by the histopathology grade of pancreatic fibrosis and presence of pancreatic fibrosis signs. The Kolmogorov–Smirnov test was used for the distribution type definition of the obtained data. For non-Gaussian distributed data, non-parametric criteria for data analysis were used. The statistical significance of intergroup differences of the independent samples was assessed by the Kruskal–Wallis test, median test, Jonckheere–Terpstra test, and Mann–Whitney U-test. *p*-values of <0.05 were considered statistically significant, and a 95% confidence interval was used. Correlations between MDCT, biomarkers, and histopathology indicators were analyzed using Spearman’s correlation coefficient (rho). Statistical analysis was performed using a dedicated statistical analysis software package (SPSS Statistics, version 23; IBM, Armonk, NY, USA).

## 3. Results

Of the 74 patients, 33 (44.6%) were male and 41 (55.4%) were female, with a mean age of 56.9 (25–84). There were 64 (86.5%) patients with PC, 8 (10.8%) patients with benign pancreatic tumors, and 2 (2.7%) patients with severe CP. The tumor localization and surgery types are presented in Table 3 and Table 4.

The patient distribution based on pancreatic fibrosis grade and signs before intergroup difference analysis is presented in Table 5 and Table 6.

### 3.1. MDCT Post-Processing Results: Intergroup Differences

We assessed intergroup differences between MDCT post-processing values (non-enhanced images, PP and VP, NCER PP, NCER VP, and CER) based on the histopathology grade of pancreatic fibrosis and presence of pancreatic fibrosis signs. We found statistically significant (*p* < 0.05) intergroup differences between unenhanced pancreas density, mean NCER VP, and CER depending on pancreatic fibrosis grades and the presence of inflammation and pancreatic duct epithelium metaplasia. The difference in unenhanced pancreas density depending on the presence of peripheral nerves reported by histopathology analysis was also statistically significant (*p* < 0.05). The data on MDCT post-processing values by pancreatic fibrosis grades and fibrosis signs are summarized in Table 7.

### 3.2. Biomarker Levels: Intergroup Differences

We found statistically significant intergroup differences between the mean values of FN in groups divided by intralobular fibrosis grade (*p* = 0.009) and the integrative index of fibrosis (*p* = 0.02). Concentrations of FN decreased significantly with the severity of the fibrosis according to the integrative index of fibrosis. Blood serum HA levels were significantly lower in the presence of inflammation in pancreatic tissue (90.2 vs. 74.2, *p* = 0.04) and higher in the presence of prominent and enlarged peripheral nerves (62.4 vs. 132.3, *p* = 0.02). We did not find any significant intergroup differences between mean values of blood serum MMP-1 and MMP-9 level within all histopathology criteria (*p* > 0.05).

The data on the laboratory parameters in different pancreatic fibrosis groups are presented in Table 8.

### 3.3. Correlation Analysis

Unenhanced pancreas density value and CER showed a statistically significant correlation with peri- and intralobular fibrosis grade, integrative index of fibrosis, inflammation in pancreatic tissue, and pancreatic duct epithelium metaplasia, while NCER VP significantly correlated with perilobular fibrosis grade, integrative fibrosis index, and inflammation only (*p* < 0.05).

The blood serum FN level showed a weak negative correlation with intralobular fibrosis grade (rho = −0.31, *p* = 0.008). The blood serum level of HA positively correlated with the presence of prominent and enlarged peripheral nerves (rho = 0.23, *p* = 0.05). In addition, we revealed a negative correlation between blood serum HA level and unenhanced pancreas density value (rho = −0.42, *p* = 0.0001). The correlation of MMP-1 and MMP-9 with histopathology pancreatic fibrosis grade and signs did not reach statistical significance. However, we did not see any strong or moderate correlations in our results. The strongest of the obtained correlation values was that between inflammation in pancreatic tissue and the mentioned MDCT post-processing indicators. The results of the correlation analysis are summarized in Table 9.

## 4. Discussion

The main feature of our study was comparing MDCT results and biomarkers’ levels with histopathology markers of pancreatic fibrosis signs in pancreatic tissue. The study has demonstrated that the levels of some MDCT post-processing indicators (unenhanced pancreas density, NCER VP, and CER values) and biomarkers (FN and HA) in blood serum correlate with grades of pancreatic fibrosis and, thus, might be used in its early diagnosis.

The degree of pancreatic tissue attenuation or density depends on the attenuation characteristics of the tissue components and microcirculation [34]. Pancreatic tissue attenuation was used by Ohgi and Sano et al. for non-invasive pancreatic fistula prediction [35,36]. These Japanese researchers showed that low CT attenuation of the pancreas might represent fibrosis of the pancreas and a decrease in pancreatic acinar cells. In our study, the unenhanced pancreatic density in moderate and severe pancreatic fibrosis was significantly lower than that in mild pancreatic fibrosis (32.3 HU and 33.9 HU vs. 40.7 HU, respectively; *p* = 0.007). In addition, there was a statistically significant negative correlation between unenhanced pancreatic tissue density and the integrative fibrosis index (rho = −0.34, *p* = 0.003). There were no significant differences between the pancreatic tissue attenuation in contrast-enhanced phases and any histopathology criteria (*p* > 0.05). This could be explained by the leveling effect of contrast enhancement because it can be affected by the contrast media amount and injection rate, as well as imaging time after contrast administration, as was also mentioned by Ohgi et al. [35].

Special post-processing approaches are essential for pancreatic fibrosis analysis with MDCT as routine MDCT cannot give credible and sufficient information about pancreatic fibrosis [37]. Hashimoto et al. presented data showing that the pancreatic late/early attenuation ratio was positively correlated with pancreatic fibrosis in their retrospective study [32]. They evaluated the predictive value of the pancreatic CT enhancement pattern indicating pancreatic fibrosis. In our study, we used a similar approach to calculate this ratio, CER. The CER was significantly higher in severe pancreatic fibrosis than in mild pancreatic fibrosis (mean CER = 2.89 vs. 0.76, respectively; *p* = 0.008). Moreover, we saw a significant positive correlation between the CER and grades of perilobular fibrosis (rho = 0.38, *p* = 0.001). These results are concordant with the data published by Hashimoto et al. [32].

In our study, we also used a method described by Torphy et al. [31]. We calculated the NCER for two phases of contrast enhancement, PP and VP, and only NCER VP showed significant intergroup differences. In particular, we saw significant moderate growth of NCER VP in severe compared with mild pancreatic fibrosis (mean NCER VP 0.62 vs. 0.47, respectively; *p* = 0.006). Torphy et al. also demonstrated higher NCER VP in tumors with high stromal density. Additionally, they saw no significant differences in NCER PP in tumors with high or low stromal density. We found no contradictions with the results in this paper, so it shows the feasibility of this approach for non-invasive analysis of stroma.

We also found that pancreatic tissue inflammation as a sign of CP and fibrotic disorder in the pancreas significantly (*p* < 0.05) affected both biomarker levels and MDCT post-processing indicators. Unfortunately, we did not find any studies similar to this in the published literature. The vast majority of articles closely related to the topic are concerning acute pancreatitis, which has a different MDCT pattern to CP; thus, we could not use those data in this pancreatic fibrosis study.

We noted intergroup differences in the blood serum FN level in groups divided by intralobular fibrosis grade (*p* = 0.009) and the integrative index of fibrosis (*p* = 0.02). The FN level was higher in mild compared with severe pancreatic fibrosis (mean 104.4 vs. 68.9 μg/mL). This might be because FN is secreted by pancreatic stellate cells at all stages of CP, but these cells’ activity is higher during early fibrotic changes [19,38]. Moreover, our results agree with experimental studies of CP pathophysiology conducted in rodent models [22,39].

What remains unclear, however, is the relatively high mean FN level at grade “0” of perilobular fibrosis. That leaves space for future research regarding how FN could be implicated in pancreatic fibrosis diagnosing and CP staging and stratifying.

In our study, we saw no significant relations between blood serum HA level and pancreatic fibrosis grade as was shown in liver diseases with fibrosis [23]. Nevertheless, we found significantly lower blood serum HA levels in specimens with histopathology signs of inflammation than in the specimens without those (90.2 vs. 74.2 ng/mL; *p* = 0.04). These results contradict experimental data where authors described increasing blood serum HA levels in models with induced CP [22]. In addition, we demonstrated a significant (*p* = 0.02) increase in HA level in specimens with prominent and enlarged peripheral nerves in resected samples. We did not find any information regarding HA’s role in nerves distribution in the pancreas, but this phenomenon might be explained by one of the biological roles of HA, namely its impact on nerve overgrowth and proliferation [40,41]. These different ways of HA blood serum concentration changing could be the cause of the absence of any significant connection between blood serum HA level and pancreatic fibrosis grades.

MMP-1 and MMP-9 in blood serum were considered promising markers at our study conception and planning phases. Unfortunately, we did not find any significant intergroup differences between mean values of blood serum MMP-1 and MMP-9 levels within all histopathology criteria (*p* > 0.05). This fact contradicts studies that have shown an MMP-9 increase in patients with CP [18,42]. Venkateshwari et al. showed that patients with CP had higher serum concentrations of MMP-9 than healthy individuals (18.325 ± 3.023 vs. 13.621 ± 0.5978 ng/mL). Our results are concordant with the results of Yokota et al., who revealed no significant signs of MMP-9 expression in models with induced pancreatitis [19]. We also did not see an MMP-1 decrease in patients with CP, as was described in some studies [22,43].

Our study has a number of potential limitations. Firstly, the study population was composed of a relatively small number of patients from a single institution. The small number of patients selected for participation could, however, be considered a representative PC cohort as the obtaining of specimens in CP must be medically justified and indicated; even so, it is still linked with a high complication rate, so only patients with preplanned pancreatic resection were included. Moreover, it was the reason for the impossibility of analysis differences between pancreatic fibrosis development in patients with different pancreatic diseases. The use of a single center in this setting might be addressed in the future by extending the study group to a multicenter cohort.

Secondly, not all the existing MDCT post-processing or the histochemistry and immunohistochemistry tests that are potentially effective in pancreatic fibrosis diagnosis were performed as the published research data on the topic are limited. Given the acquired results, the upcoming analysis of the extended cohort will include new variants of MDCT post-processing based on attenuation measurement and contrast enhancement dynamics along with a widened spectrum of histopathology examinations.

Notwithstanding these limitations, the study suggests that there are parameters of MDCT post-processing and biomarkers (fibronectin and hyaluronic acid) that correlate and might be sensible in diagnosing pancreatic fibrosis and could work as a non-invasive diagnostic marker.

## 5. Conclusions

In this study, we sought to find a correlation between MDCT post-processing values including pancreatic density in non-enhanced images, PP and VP, NCER PP, NCER VP, and CER; and mean values of FN, HA, MMP-1, and MMP-9 in groups based on the histopathology grade of pancreatic fibrosis and presence of pancreatic fibrosis signs. The preliminary results of our study showed that among the MDCT post-processing indicators, based on contrast enhancement dynamics analysis, NCER VP and CER were increased in patients with severe pancreatic fibrosis. Unenhanced pancreas density can provide additional information due to its decline as pancreatic fibrosis progresses. The blood serum FN level decreased in patients with severe pancreatic fibrosis and was higher in patients with mild pancreatic fibrosis. HA decreased in patients with histopathological signs of inflammation and increased in patients with enlarged peripheral nerves in resected samples. Despite the specimens not being obtained in early CP itself and the data on pancreatic fibrosis in general and its non-invasive diagnostics in particular being limited, the demonstrated results suggest the feasibility of some MDCT post-processing indicator (NCER, CER, and unenhanced pancreas density value) and serum biomarker (FN and HA) levels for early diagnosis of pancreatic fibrosis.

## Figures and Tables

**Figure 1 diagnostics-13-02435-f001:**
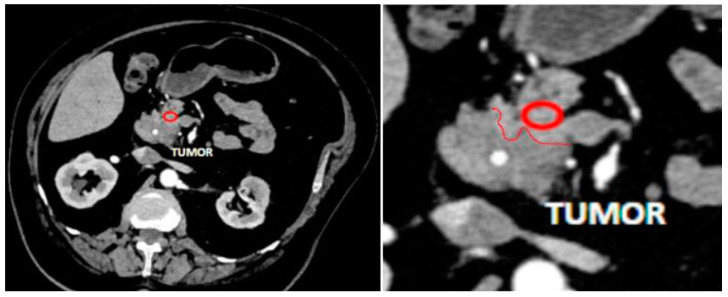
An example of measurement of pancreatic tissue near a tumor. MDCT in arterial phase, axial view. Red circle indicates a region of measurement, red line is a border between tumor and intact pancreatic tissue.

**Figure 2 diagnostics-13-02435-f002:**
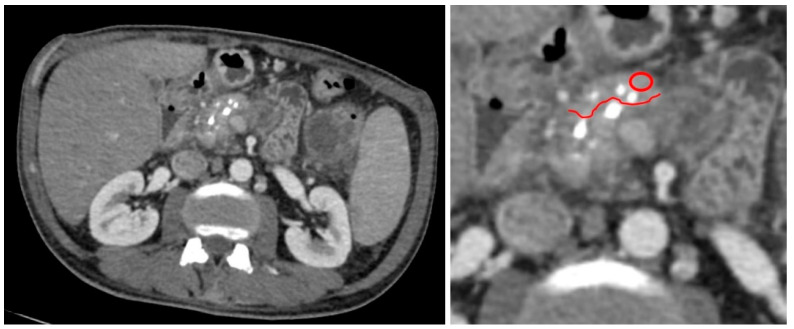
An example of measurement of pancreatic tissue in the part that would be resected during surgery in patient with CP. MDCT in arterial phase, axial view. Red circle indicates a region of measurement, red line is a border of tissue which was preplanned to be resected along surgery.

**Figure 3 diagnostics-13-02435-f003:**
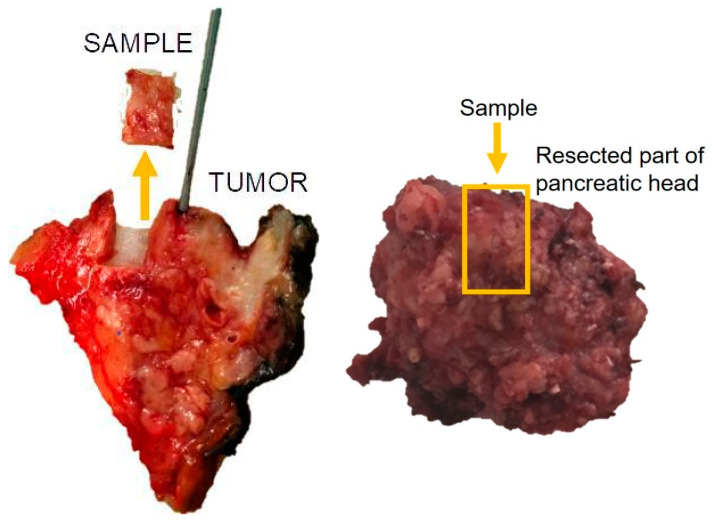
An example of resection of a sample for pancreatic fibrosis analysis.

**Table 1 diagnostics-13-02435-t001:** Formulae used in MDCT results post-processing.

Value	Formula
NCER during the PP	(Pancreatic density in PP − Pancreatic density in precontrast phase)/ (Blood density in aorta in PP − Blood density in aorta in precontrast phase)
NCER during the VP	(Pancreatic density in VP − Pancreatic density in precontrast phase)/ (Blood density in aorta in VP − Blood density in aorta in precontrast phase)
CER	(Pancreatic density in VP − Pancreatic density in precontrast phase)/(Pancreatic density in PP − Pancreatic density in precontrast phase)

Abbreviations: NCER—normalized contrast enhancement ratio; PP—pancreatic phase; VP—venous phase; CER—contrast enhancement ratio.

**Table 2 diagnostics-13-02435-t002:** Grade of fibrosis according to Kloppel and Maillet adapted from [33].

Fibrosis Patterns	Fibrosis Degree		
	Mild	Moderate	Severe
Perilobular fibrosis			
Focal	1	2	3
Diffuse	4	5	6
Intralobular fibrosis			
Focal	1	2	3
Diffuse	4	5	6
Integrative index	Mild fibrosis		≤6
	Moderate fibrosis		7–9
	Severe fibrosis		10–12

**Table 3 diagnostics-13-02435-t003:** Localization of tumors in PC patients.

Localization	*N*	%
Uncinate process	13	18.1
Head	38	52.8
Isthmus	4	5.5
Body	6	8.3
Tail	11	15.3
Total	72	100

**Table 4 diagnostics-13-02435-t004:** Surgery types performed in patients.

Type of Surgery	*N*	%
Pancreaticoduodenectomy	48	64.9
Distal pancreatectomy	18	24.3
Total pancreatectomy	6	8.1
Drainage surgery	2	2.7
Total	74	100

**Table 5 diagnostics-13-02435-t005:** Patient distribution according to pancreatic fibrosis grade.

Pancreatic Fibrosis Grade	*N*	%
Perilobular fibrosis grade		
0	1	1.4
1	18	24.3
2	11	14.9
3	3	4
4	12	16.2
5	20	27
6	9	12.2
Total	74	100
Intralobular fibrosis grade		
0	3	4
1	25	33.8
2	8	10.8
3	2	2.7
4	17	23
5	13	17.6
6	6	8.1
Total	74	100
Integrative index of fibrosis		
Mild	24	32.4
Moderate	9	12.2
Severe	41	55.4
Total	74	100

**Table 6 diagnostics-13-02435-t006:** Patient distribution according to pancreatic fibrosis signs.

Pancreatic Fibrosis Sign	*N*	%
Inflammation		
No	32	43.2
Yes	42	56.8
Total	74	100
Pancreatic duct epithelium metaplasia		
No	34	45.9
Yes	40	54.1
Total	74	100
Peripheral nerves		
No	51	68.9
Yes	23	31.1
Total	74	100
Protein plugs		
No	40	54.1
Yes	34	45.9
Total	74	100

**Table 7 diagnostics-13-02435-t007:** MDCT post-processing values in pancreatic fibrosis grade groups.

	Unenhanced Pancreas Density Mean Values, HU	*p*	PP Pancreas Density Mean Values, HU	*p*	VP Pancreas Density Mean Values, HU	*p*	Mean NCER PP	*p*	Mean NCER VP	*p*	Mean CER	*p*
**Pancreatic fibrosis grade**												
Perilobular fibrosis grade												
0	55	**0.01**	102	0.76	110	0.18	0.17	0.8	0.59	**0.005**	1.17	**0.003**
1	39.6		94.1		80		0.36		0.45		0.73	
2	37		93		77.9		0.29		0.54		1.08	
3	40.3		95.3		94		0.24		0.54		1.15	
4	34.1		102.4		89.1		0.33		0.59		0.82	
5	31.7		88.1		82.3		0.33		0.64		5.17	
6	33.75		85		76.3		0.33		0.59		1.27	
Intralobular fibrosis grade												
0	46.7	**0.01**	115.3	0.12	98	0.058	0.24	0.93	0.53	**0.012**	0.8	**0.05**
1	39.2		90.9		78.5		0.31		0.49		0.77	
2	31		83.9		79.6		0.39		0.57		9.47	
3	34		77.5		90		0.25		0.57		1.42	
4	36.2		99.8		91.9		0.34		0.59		0.99	
5	29.6		88.3		79.3		0.37		0.68		0.81	
6	36.8		68		70.3		0.24		0.54		1.5	
	**Unenhanced pancreas density mean values, HU**	** *p* **	**PP pancreas density mean values, HU**	** *p* **	**VP pancreas density mean values, HU**	** *p* **	**Mean** **NCER PP**	** *p* **	**Mean** **NCER VP**	** *p* **	**Mean CER**	** *p* **
Integrative index of fibrosis												
Mild	40.7	**0.007**	92.1	0.9	80.5	0.06	0.3	0.56	0.47	**0.006**	0.76	**0.007**
Moderate	32.2		91.5		74.9		0.38		0.53		1.03	
Severe	33.9		89.4		85		0.33		0.62		2.89	
**Pancreatic fibrosis sign**												
Inflammation												
No	39.5	**0.004**	94.3	0.29	80	0.19	0.34	0.46	0.49	**0.007**	0.73	**0.002**
Yes	33.4		87.7		84.1		0.32		0.62		2.94	
Pancreatic duct epithelium metaplasia												
No	39.2	**0.04**	89.7	0.11	83.9	0.48	0.31	0.72	0.54	0.68	0.76	**0.007**
Yes	33.6		91.6		80.9		0.34		0.56		2.8	
Peripheral nerves												
No	37.1	**0.03**	93.1	0.52	83.6	0.77	0.33	0.68	0.55	0.2	2.2	0.16
Yes	30.99		84.8		79.4		0.31		0.54		1	
Protein plugs												
No	36.4	0.22	89.6	0.65	80.6	0.34	0.3	0.25	0.53	0.79	0.9	0.14
Yes	33.6		91.6		84.3		0.36		0.56		3	

Abbreviations: MDCT—multidetector computed tomography; NCER—normalized contrast enhancement ratio; PP—pancreatic phase; VP—venous phase; CER—contrast enhancement ratio. *p*-values of statistically significant differences are highlighted in bold.

**Table 8 diagnostics-13-02435-t008:** Intergroup differences in fibrosis biomarker levels by pancreatic fibrosis grades and signs groups.

Parameter	FN Mean Values, μg/mL	*p*	HA Mean Values, ng/mL	*p*	MMP-1, Mean Values, ng/mL	*p*	MMP-9, Mean Values, ng/mL	*p*
**Pancreatic fibrosis grade**								
Perilobular fibrosis grade								
0	132	0.13	17.6	0.66	4.14	0.38	626.8	0.7
1	107.1		80.7		57.84		1057.3	
2	106.8		87.5		42.37		846.9	
3	43.7		40.9		68.98		1719.6	
4	85.7		56.9		56.47		1142.9	
5	79		86.3		48.97		864.8	
6	68		102.7		46.25		616.2	
Intralobular fibrosis grade								
0	93.3	**0.009**	21.1	0.16	43.9	0.5	1190.1	0.89
1	109.2		85.9		52.5		853.7	
2	80.9		55.6		51.6		1040.4	
3	47.5		36.2		69.5		965.5	
4	65.4		54.6		55.1		1164.7	
5	70.2		150.3		39.8		859.7	
6	75.2		46.1		56.9		757.8	
Integrative index of fibrosis								
Mild	104.4	**0.02**	86.5	0.8	49.9	0.45	971.6	0.94
Moderate	95		52.9		60.6		901.7	
Severe	68.9		81.4		49.9		957.4	
**Pancreatic fibrosis sign**								
Inflammation								
No	101.1	0.11	90.2	**0.04**	49.8	0.56	826	0.17
Yes	79.8		74.2		52.3		1053.7	
Pancreatic duct epithelium metaplasia								
No	86.7	0.48	53.7	0.38	51.3	0.9	1013.9	0.8
Yes	90.9		101.6		51.2		905.5	
	**FN mean values, μg/mL**	** *p* **	**HA mean values, ng/mL**	** *p* **	**MMP-1, mean values, ng/mL**	** *p* **	**MMP-9, mean values, ng/mL**	** *p* **
Peripheral nerves								
No	92.4	0.48	62.4	**0.02**	52.2	0.66	983.6	0.72
Yes	81.6		132.3		49.2		905.5	
Protein plugs								
No	94.7	0.08	66.7	0.67	52.9	0.5	946.9	0.96
Yes	82.3		94.8		49.3		965.1	

Abbreviations: FN—fibronectin; HA—hyaluronic acid; MMP—matrix metalloproteinase. *p*-values of statistically significant differences are highlighted in bold.

**Table 9 diagnostics-13-02435-t009:** Correlation analysis between MDCT post-processing parameters, fibrosis biomarkers, and histopathology pancreatic fibrosis grade and signs.

Parameter	Unenhanced Pancreas Density, rho, *p*	PP Pancreas Density, rho, *p*	VP Pancreas Density, rho, *p*	NCER PP, rho, *p*	NCER VP, rho, *p*	CER, rho, *p*	FN, rho, *p*	HA, rho, *p*	MMP-1, rho, *p*	MMP-9, rho, *p*
Perilobular fibrosis grade	−0.332, 0.004	−0.176, 0.13	0.051, 0.66	−0.015, 0.9	0.283, 0.015	0.383, 0.001	−0.197, 0.09	0.047, 0.69	−0.063, 0.59	−0.144, 0.22
Intralobular fibrosis grade	−0.309, 0.007	−0.145, 0.22	0.041, 0.73	0.015, 0.9	0.218, 0.062	0.293, 0.01	−0.306, 0.008	0.17, 0.15	−0.042, 0.7	0.003, 0.98
Integrative index of fibrosis	−0.341, 0.003	−0.047, 0.69	0.239, 0.4	0.012, 0.92	0.372, 0.001	0.363, 0.001	−0.358, 0.002	0.058, 0.62	−0.011, 0.9	−0.029, 0.8
Inflammation	−0.363, 0.001	−0.125, 0.29	0.154, 0.19	−0.088, 0.5	0.311, 0.007	0.388, 0.001	−0.187, 0.11	0.114, 0.3	0.068, 0.56	0.16, 0.17
Pancreatic duct epithelium metaplasia	−0.239, 0.04	−0.188, 0.11	−0.083, 0.48	0.043, 0.7	0.076, 0.52	0.284, 0.014	0.08, 0.5	0.103, 0.38	0.013, 0.9	0.029, 0.8
Peripheral nerves	−0.247, 0.034	−0.076, 0.52	−0.035, 0.77	−0.049, 0.7	−0.034, 0.78	0.165, 0.16	−0.081, 0.5	0.225, 0.05	−0.051, 0.66	0.042, 0.72
Protein plugs	−0.144, 0.22	0.053, 0.66	0.107, 0.36	0.135, 0.25	0.198, 0.09	0.174, 0.14	−0.204, 0.08	0.05, 0.67	−0.078, 0.51	−0.006, 0.96

Abbreviations: MDCT—multidetector computed tomography; NCER—normalized contrast enhancement ratio; PP—pancreatic phase; VP—venous phase; CER—contrast enhancement ratio; FN—fibronectin; HA—hyaluronic acid; MMP—matrix metalloproteinase. Statistically significant correlations are highlighted in bold.

## Data Availability

The data are available upon request to the corresponding author.

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
