# Peer review of "Contrast-Enhanced Computed Tomography and Laboratory Parameters as Non-Invasive Diagnostic Markers of Pancreatic Fibrosis"

_diagnostics, 2023, doi:10.3390/diagnostics13142435_

Round 1
Reviewer 1 Report
With the present work the authors aim to investigate a relationship between pancreatic fibrosis and the results of multidetector computed tomography post-processing and laboratory parameters.
It is a well-designed and conducted study with limitations inherent to the collection of pathological samples.
In the abstract several acronyms are used before their written representation, which makes it difficult to understand.
Author Response
Point: In the abstract several acronyms are used before their written representation, which makes it difficult to understand.
Response: We appreciate the reviewer for the manuscript evaluation and the comment. We have added the written representations for the acronyms in the abstract.
Reviewer 2 Report
This manuscript submitted by Igor Khatkov using multidetector computed tomography (MDCT) in combination with several blood markers of fibronectin (FN), hyaluronic acid (HA), matrix metalloproteinase (MMP)-1, and MMP-9 plasma levels to assess pancreatic fibrosis (PF) in CP or PC tissues. This information is useful for improving early diagnosis of PF. Some comments:
Major
1. The abstract, especially the first sentence reads as this is a study about CP only.
2. Both CP and PC patients were recruited in this study, but there is no analysis to show how PF occurring differently/indifferently between the two diseases and how the defined biomarker performing in CP and PC respectively.
Minor
3. Other than that, I have one minor comments on the confusion of what blood sample, serum or plasma, was used for what test, please clarify.
4. Figure 1 and figure 2: example of measurement for CP should be included.
Author Response
We appreciate the reviewer’s work on our manuscript. We present answers on comments below.
Major point 1: The abstract, especially the first sentence reads as this is a study about CP only.
Response 1: We are very grateful for reviewer’s kind comment. Our study is dedicated to early diagnosis of pancreatic fibrosis, which is a part of most pancreatic disorders and might be reversed if early detected. We changed abstract to clarify an object of our study.
Major point 2: Both CP and PC patients were recruited in this study, but there is no analysis to show how PF occurring differently/indifferently between the two diseases and how the defined biomarker performing in CP and PC respectively.
Response 2: We thank reviewer for the comment. As covered before, the only way to receive specimens adequate for histopathology analysis is pancreatic surgery, that is mandatory in patients with pancreatic cancer (PC), majority benign pancreatic tumors and severe chronic pancreatitis (CP). In our study, only two patients with CP preplanned for surgery were included. Therefore, we cannot analyze differences between pancreatic fibrosis development in patients with CP and PC. We plan to include more patients with chronic pancreatitis who need pancreatic surgery in this study to clarify features of pancreatic fibrosis in both CP and PC.
Minor point 1: Other than that, I have one minor comments on the confusion of what blood sample, serum or plasma, was used for what test, please clarify.
Response 1: We thank reviewer for the comment. We used serum blood sample for all biomarker tests. We clarified it in updated manuscript.
Minor point 2: Figure 1 and figure 2: example of measurement for CP should be included.
Response 2: We added additional examples of measurements for CP in figures.